# Proteome constraints reveal targets for improving microbial fitness in nutrient-rich environments

Yu Chen[1,2,†] , Eunice van Pelt-KleinJan[3,4,†] , Berdien van Olst[3,5,6] , Sieze Douwenga[3,4] ,
Sjef Boeren[3,6] , Herwig Bachmann[3,4,7] , Douwe Molenaar[3,4] , Jens Nielsen[1,2,8,9,*] &
Bas Teusink[3,4,**]

## Abstract

Cells adapt to different conditions via gene expression that tunes metabolism for maximal fitness. Constraints on cellular proteome may limit such expression strategies and introduce trade-offs. Resource allocation under proteome constraints has explained regulatory strategies in bacteria. It is unclear, however, to what extent these constraints can predict evolutionary changes, especially for microorganisms that evolved under nutrient-rich conditions, i.e., multiple available nitrogen sources, such as *Lactococcus lactis*. Here, we present a proteome-constrained genome-scale metabolic model of *L. lactis* (pcLactis) to interpret growth on multiple nutrients. Through integration of proteomics and flux data, in glucose-limited chemostats, the model predicted glucose and arginine uptake as dominant constraints at low growth rates. Indeed, glucose and arginine catabolism were found upregulated in evolved mutants. At high growth rates, pcLactis correctly predicted the observed shutdown of arginine catabolism because limited proteome availability favored lactate for ATP production. Thus, our model-based analysis is able to identify and explain the proteome constraints that limit growth rate in nutrient-rich environments and thus form targets of fitness improvement.

**Keywords** ccpA; laboratory evolution; *Lactococcus lactis*; metabolic modeling; proteome constraint

**Subject Categories** Metabolism; Microbiology, Virology & Host Pathogen Interaction; Proteomics

**Mol Syst Biol. (2021) 17: e10093**

## Introduction

The fitness of unicellular organisms is determined by adaptions to environmental conditions (Goel *et al*, 2012b) and is optimized by regulating metabolic processes that generally lead to higher growth rates (Chubukov *et al*, 2014). Growth rates are finite, as the metabolic processes supporting growth are constrained through limits imposed by external conditions, e.g., nutrient availability, and internal factors that relate to cell morphology, enzyme kinetics, and physicochemical properties such as solvent capacities. In particular, constraints on the allocation of the proteome, due to limited membrane area or intracellular volume, have aided to understand metabolic adaptions of microorganisms (Beg *et al*, 2007; Molenaar *et al*, 2009; de Groot *et al*, 2020), specifically, the overflow metabolism in *Escherichia coli* (Zhuang *et al*, 2011; O'Brien *et al*, 2013; Basan *et al*, 2015) and the Crabtree effect in *Saccharomyces cerevisiae* (Nilsson & Nielsen, 2016; Sánchez *et al*, 2017; Chen & Nielsen, 2019).

However, much less is known about metabolic adaptations in other organisms, and especially those cultivated under conditions with multiple available substrates, such as in nutrient-rich environments like the gut or food. It was previously shown that anaerobic "overflow" metabolism in the lactic acid bacterium *Lactococcus lactis*, i.e., the transition from energy-efficient mixed acid fermentation to less energy-efficient lactic acid fermentation, is not accompanied by changes in associated protein levels (Goel *et al*, 2015), questioning the generality of the resource allocation paradigm. However, consistent changes in both gene expression and metabolic levels were observed in amino acid metabolism, prompting us to revisit the cellular economics of *L. lactis*.

*L. lactis* is an important model lactic acid bacterium and work horse for the dairy industry—production of cheese in particular (Papadimitriou *et al*, 2016; Kok *et al*, 2017). Many strains are

---

1   Department of Biology and Biological Engineering, Chalmers University of Technology, Gothenburg, Sweden
2   Novo Nordisk Foundation Center for Biosustainability, Chalmers University of Technology, Gothenburg, Sweden
3   TiFN, Wageningen, the Netherlands
4   Systems Biology Lab, Amsterdam Institute of Molecular and Life Sciences (AIMMS), Vrije Universiteit Amsterdam, Amsterdam, The Netherlands
5   Host-Microbe Interactomics, Wageningen University & Research, Wageningen, The Netherlands
6   Laboratory of Biochemistry, Wageningen University & Research, Wageningen, The Netherlands
7   NIZO Food Research, Ede, The Netherlands
8   Novo Nordisk Foundation Center for Biosustainability, Technical University of Denmark, Lyngby, Denmark
9   BioInnovation Institute, Copenhagen N, Denmark
    *Corresponding author. Tel: +46 317723804; E-mail: nielsenj@chalmers.se
    **Corresponding author. Tel: +31 205989435; E-mail: b.teusink@vu.nl
    †These authors contributed equally to this work

auxotrophic for at least five amino acids (Jensen & Hammer, 1993; Teusink & Molenaar, 2017), and thus, *L. lactis* strains are grown in nutrient-rich environments where amino acids are usually in excess. Many of these amino acids participate not only in anabolic processes but also in catabolism to contribute to energy metabolism (Fernández & Zúñiga, 2006). For example, arginine catabolism directly yields ATP and is tightly regulated (Crow & Thomas, 1982), while other amino acids can contribute to pH or redox homeostasis, saving ATP-costly alternatives (Novák & Loubiere, 2000). It is, however, not known whether proteome constraints guide choices in non-sugar substrates, e.g., amino acids.

# Results

### Construction and evaluation of pcLactis

We therefore developed pcLactis, a proteome-constrained genome-scale metabolic model of *L. lactis*. We first updated a published genome-scale metabolic model for *L. lactis* MG1363 (Verouden *et al*, 2009; Flahaut *et al*, 2013) mostly by adding transport capacities and gene-protein-reaction associations (Dataset EV1). Second, we added the gene expression processes, including transcription, stable RNA cleavage, mRNA degradation, tRNA modification, rRNA modification, tRNA charging, ribosomal assembly, translation, protein maturation and assembly, and protein degradation (Fig 1A). By integrating the two reconstructions, we obtained pcLactis, which accounts for 725 protein-coding genes and 81 RNA genes in *L. lactis* MG1363. According to the PaxDb database (Wang *et al*, 2015), pcLactis accounts for approximately 60% of the total proteome by mass (Dataset EV2). The model's proteome therefore includes 40% unmodeled protein of average amino acid composition. We constrain each metabolic flux at the maximal rate of the associated enzyme, which is a function of the enzyme concentration and turnover rate. Thus, we compute minimal enzyme levels to sustain metabolic flux. Enzyme levels follow from mass balancing synthesis rate and degradation and dilution by growth (Fig 1B). Total protein synthesis rates are constrained by ribosomal translation capacity and by a so-called total proteome constraint, i.e., a maximum size of the proteome. We use inactive enzyme, again with average amino acid composition, to fill up the total proteome in cases of low enzymatic activity, e.g., at low growth rates. It should be noted that the inactive enzyme does not represent any specific enzyme but just accounts for the sum of inactive fractions of all undersaturated enzymes. Therefore, the inactive enzyme can be seen as the excess capacity of the total modeled proteome, and the constraint of the total modeled proteome is reached if the inactive enzyme is zero.

We used published proteomics and flux data from chemostats for model evaluation, and simulated glucose-limited conditions by minimizing the glucose concentration at a fixed specific growth rate with an upper bound on the expression of the glucose transporter. The model predicted the glucose uptake rates well (Fig 1C) and indicated—as expected—increased saturation of the glucose transporter with growth rate (Fig 1D). Additionally, pcLactis predicted a metabolic switch from mixed acid to lactic acid fermentation, with corresponding changes at the proteome level (Appendix Fig S1A), but at a much higher growth rate than experimentally observed (Fig 1C). However, experimental evidence showed that this metabolic switch

in *L. lactis* does not relate to considerable proteome changes (Goel *et al*, 2015) and therefore the switch predicted by pcLactis probably does not reflect the true reason for this change in glucose metabolism. More importantly, pcLactis predicted that at a dilution rate (D) higher than 0.5/h, the fraction of inactive enzyme becomes zero (Fig 1E). In the model, this means that all available proteome space is being actively used for metabolic function and that under such conditions any flux change can only be brought about by changes in protein levels, not enzyme saturation. This conclusion is supported by the fact that most glycolytic enzymes reach the highest saturation above D = 0.5/h (Goel *et al*, 2015) and that proteomics data showed genome-wide protein reallocation when growth rate increased beyond that point (Appendix Fig S2).

To identify the "active" constraints, i.e., constraints that limit the growth rate in pcLactis, we applied a sensitivity analysis on the proteome-constrained model. Glucose transport expression was an active constraint at low growth rates and its sensitivity dropped at the moment the inactive enzyme reached zero (Fig 1F). At that point, the total proteome also became limiting. This reflects the increased demand for proteome resource at high growth rates, both for metabolic fluxes and protein translation machinery (Appendix Fig S1A). Thus, around D = 0.5/h, the model switched from glucose-limited to combined glucose and proteome–limited.

### pcLactis identifies arginine uptake as the most-active constraint on growth and captures arginine shift

In the model, the transition to proteome-limited growth was reflected in amino acid metabolism, not in glucose metabolism. However, including amino acid uptake in pcLactis was challenging as the amino acid consumption could not be constrained by the model due to insufficient data of expression and kinetics for amino acid transport systems. This would lead to an overestimation of uptake and growth rate when setting free exchange rates of all 20 amino acids to mimic the medium of *L. lactis*, where amino acids are mostly in excess. However, published amino acid data (Goel *et al*, 2015) showed that apart from aspartate and glutamate all the detected amino acids were taken up linearly with growth rate (Fig 2 A). We therefore imposed growth rate-dependent upper bounds on the uptake rates of all amino acids based on these measurements. To distinguish anabolism and catabolism, we compared the uptake of amino acids with their tRNA charging flux, which represents the flux toward protein synthesis. To analyze the results, we performed a so-called scaled reduced cost analysis, which is a sensitivity analysis of the uptake bounds on the growth rate (Fig 2B).

The predicted uptake fluxes of most amino acids follow the upper bound set by the experimental data. Many of those amino acids were overconsumed and thus catabolized, in agreement with model predictions (Fig 2A). Based on the scaled reduced cost analysis, their catabolism contributed to growth rate (the mechanistic basis of many of those were previously analyzed for *Lactobacillus plantarum* (Teusink *et al*, 2006)). Those amino acids with little impact on growth rate, based on the scaled reduced cost analysis, are taken up by *L. lactis* according to protein synthesis demand. Thus, pcLactis can explain overconsumption of specific amino acids according to growth rate optimization. This is interesting, as many of the catabolic products contribute to flavor formation in food fermentations. Leucine, tryptophan, and to a lesser extent valine

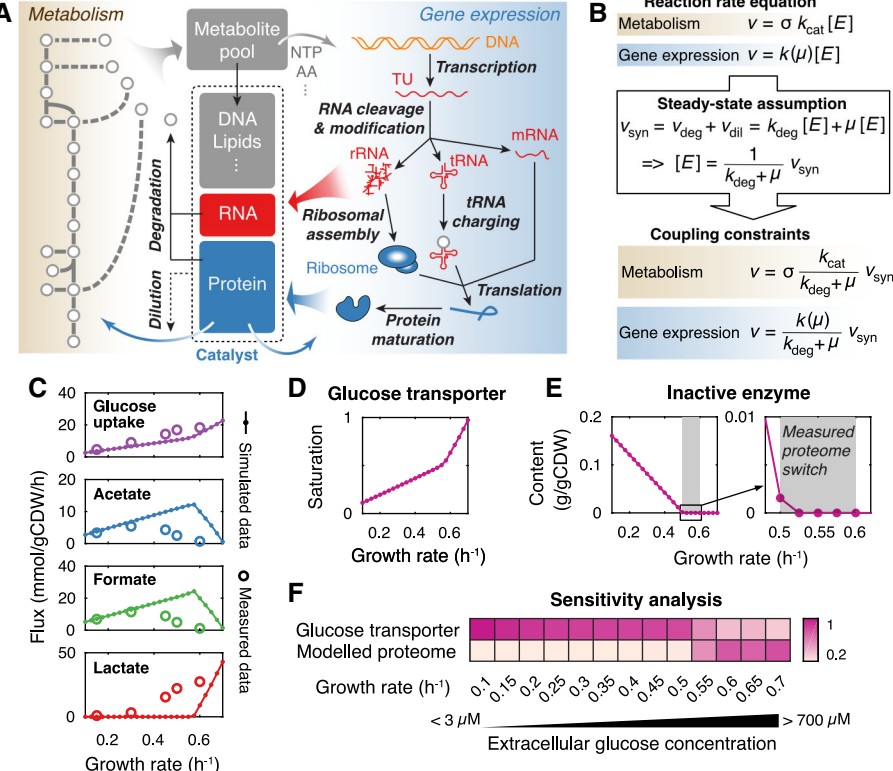

**Figure 1. Overview of pcLactis and simulations of glucose-limited conditions.**

A   The model explicitly accounts for reactions of metabolism and gene expression processes. Metabolic reactions produce metabolites and energy for not only biomass formation but also gene expression processes. The reactions of gene expression, on the other hand, synthesize RNA and proteins, which catalyze reactions of metabolism and gene expression processes as machineries or enzymes. In addition, pcLactis accounts for degradation of mRNA and proteins as well as dilution of biomass constituents during cell division.

B   Coupling constraints in pcLactis. The coupling constraint allows for relating the reaction rate to the synthesis rate of its catalyst based on the reaction rate equation and steady-state assumption, where turnover rates $k_{cat}$ of metabolic enzymes, catalytic rates $k$ of machineries, and degradation constants $k_{deg}$ of the catalysts are needed.

C   Simulated exchange fluxes compared with experimentally measured data (Goel *et al*, 2015).

D   Simulated saturation of glucose transporter.

E   Simulated inactive enzyme. The inactive enzyme is the sum of the enzymes that are synthesized but do not carry fluxes. The production of the inactive enzyme indicates that total proteome is not constrained. The gray area represents where the proteome switch occurs in experiments (Goel *et al*, 2015), which is between 0.5 and 0.6/h.

F   Sensitivity analysis for glucose transporter and modeled proteome at different growth rates. Color represents the sensitivity score. A higher score indicates a greater impact of a given increase in the constraint on growth rate.

were exceptions (Fig 2A). In *L. plantarum*, leucine and valine catabolism contributes to the redox balance, which requires a NADP-dependent glutamate dehydrogenase (Teusink *et al*, 2006); *L. lactis*, however, lacks this enzyme (Wels *et al*, 2019). The reason for overconsumption of these amino acids thus remains to be unraveled.

When the total proteome becomes an active constraint at a growth rate around 0.5/h, we observe a drop in several amino acid overconsumptions in pcLactis. This is also reflected in a drop in reduced cost of the respective amino acids (Fig 2), indicating that the metabolic benefit was not high enough for the now-constraining protein investment in the catabolic pathway. Experimentally, not all amino acids dropped at this point, although data are more variable at the highest growth rate. However, the most pronounced change in flux was observed for arginine, whose catabolism directly yields ATP. Arginine also had the highest (drop in) reduced cost. The predicted changes in the fluxes of catabolic products ornithine and

ammonium (Appendix Fig S1B), and the concomitant changes in protein levels of the pathway (Appendix Fig S1A) together with experimental observations (Crow & Thomas, 1982; Goel *et al*, 2015), confirm that pcLactis correctly captured the switch in arginine catabolism.

## Small model reproduces metabolic shifts and growth-limiting factors

We noted that the arginine switch preceded the onset of the switch from mixed acid to lactic acid fermentation in pcLactis (Appendix Fig S1C). We constructed a small model to analyze this switching behavior, expecting that protein efficiency, i.e., ATP produced per protein mass per time, was key (Chen & Nielsen, 2019). We defined three independent ATP-producing pathways, glycolysis pathway with mixed acid fermentation or with lactate

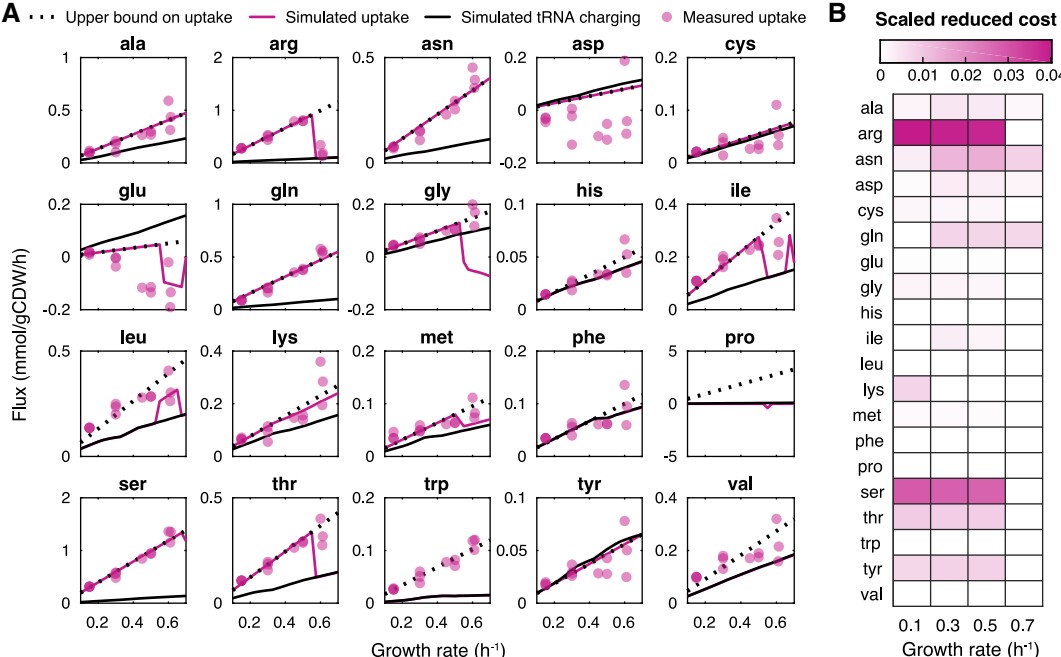

**Figure 2. Amino acid analysis with pcLactis shows importance of amino acid catabolism for growth as a function of active constraints.**

A Simulated fluxes of amino acids compared with experimentally measured data. For each amino acid, simulated fluxes of uptake and tRNA charging are displayed together with the measured uptake flux. The upper bound on the uptake is also displayed, which is the linear trendline of the experimentally measured data if the amino acid is consumed rather than being secreted. The measured data are from the study (Goel et al, 2015).

B Scaled reduced cost analysis for uptake rates of amino acids at different growth rates. Color represents the scaled reduced cost value. A higher value indicates a greater impact of a given increase in the amino acid uptake on growth rate.

formation, and arginine catabolism (Fig 3A), and estimated ATP yield, protein cost, and protein efficiency (Fig 3B) for each pathway using pcLactis. With these parameters, we formulated a linear program to maximize ATP production flux subject to total proteome constraints with flux bounds (Fig 3C). We varied glucose uptake fluxes and found three distinct phases: when the total proteome is not constrained both mixed acid fermentation and arginine catabolism are used (phase A, Fig 3D), as mixed acid fermentation has the highest ATP yield and arginine provides extra ATP at no protein burden. Once the total proteome is constrained (phase B, Fig 3D), the flux through arginine catabolism goes down due to its lowest protein efficiency. When arginine catabolism is completely inactive, mixed acid fermentation is traded in for lactate formation (phase C, Fig 3D), as its protein efficiency is lower than lactic acid fermentation. Furthermore, the small model reproduced the sensitivity results in terms of the constraints on glucose and arginine uptake (Fig 3D). Thus, the small model captures the behavior of the full model, reinforcing earlier theoretical results that the number and nature of the active constraints determine behavior, irrespective of the size of the network (de Groot et al, 2019).

## Predictions are robust to uncertainties of model parameters

The model simulations resulted in two findings. First, glucose and arginine uptake limit growth when the total proteome is not constrained. Second, the metabolic shift in arginine catabolism can be interpreted by proteome constraints, but the shift from mixed acid to lactic acid fermentation cannot. Given the fact that pcLactis relies on a huge number of parameters, many of which were not available and estimated from databases, we investigated whether uncertainties of parameters can influence the two findings.

To this end, we varied each individual parameter of pcLactis twofold (both up and down) and compared the resulting simulated growth rate with the reference. We compared a glucose-rich condition and a glucose-limited condition in which the reference growth rate is 0.5/h. Of the 845 parameters, only 24 led to variations > 1% under the glucose-rich condition (Appendix Fig S3A), meaning that uncertainties of most individual parameters have little effect on maximal growth rate simulations. When simulating the glucose-limited condition, where the total proteome is not constrained, we found—as expected—that even less (six) parameters have an impact on growth simulations (Appendix Fig S3B), only parameters acting on glucose transport—the active constraint—affected growth rate. Additionally, a large change in the modeled proteome size, such that the model becomes proteome-constrained, was required to show an effect. Accordingly, the finding that glucose and arginine uptake could limit growth is robust to the uncertainties of most parameters, e.g., turnover rates of metabolic enzymes.

The small model was used to test the robustness of our findings to overall protein cost parameters on pathway levels, rather than individual enzyme levels. We adjusted the protein costs of the pathways and upper limits on total proteome and arginine uptake, again twofold in either direction, and examined the model behavior across

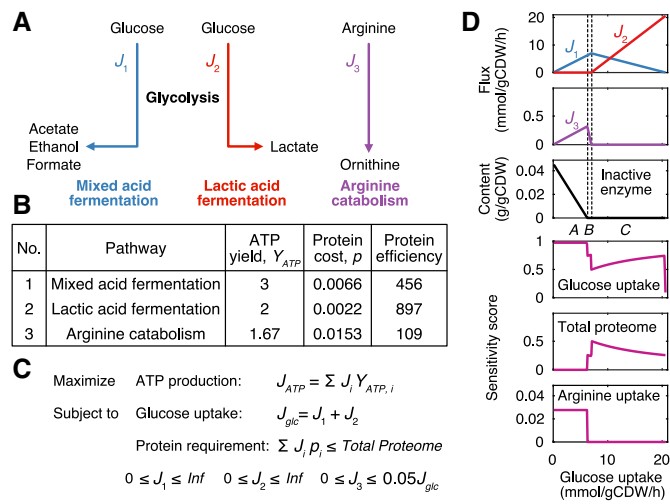

**Figure 3.  The switches between ATP yielding pathways in *L. lactis* can be explained from a protein efficiency point of view.**

A   A small model derived from pcLactis to investigate primary ATP-producing pathways in *L. lactis*. The small model consists of three independent pathways, including glycolysis pathway followed by mixed acid fermentation, glycolysis pathway followed by lactic acid fermentation, and arginine catabolism.

B   Parameters of ATP-producing pathways inferred from pcLactis. ATP yield is ATP generated per glucose (for pathway 1 and 2) or arginine (for pathway 3) consumed. Note that the ATP yield of the arginine pathway is 1.67, which is due to the fact that besides one mole of ATP produced by the carbamate kinase reaction, by degrading one mole of arginine *L. lactis* does not have to export two moles of proton, which equivalently saves 0.67 mole of ATP that should have been consumed by ATP synthase to balance protons. Protein cost is protein mass (unit: g/gCDW) required per substrate flux (unit: mmol/gCDW/h) through the pathway. Protein efficiency is ATP generated per protein mass per time with the unit of mmol of ATP per gram of protein per hour.

C   Linear programming to solve the small model. The objective function is to maximize the total ATP production flux $J_{ATP}$, which is the sum of ATP production fluxes of three pathways calculated by substrate uptake flux $J_i$ times ATP yield $Y_{ATP,i}$. The model is subject to two constraints. One is the total glucose uptake flux $J_{glc}$, which is the sum of the glucose fluxes toward pathway 1 $J_1$ and 2 $J_2$. The other is the total proteome constraint, which means that the sum of protein requirements of all pathways should be not greater than the proteome allocation in the model. For each pathway, the protein requirement is calculated by the protein cost $p_i$ times substrate flux $J_i$. In addition, there are lower and upper bounds on the uptake flux of each pathway. Specifically, $J_1$ and $J_2$ are unlimited while $J_3$ has a limited upper bound, which is assumed to be linearly correlated with total glucose uptake flux $J_{glc}$ based on experimental observation.

D   Small model simulations for a range of glucose uptake flux. The top three plots show that with increasing glucose uptake flux the decline in arginine uptake occurs firstly once the inactive enzyme disappears, and subsequently the switch from mixed to lactic acid fermentation. The inactive enzyme is the difference between the total protein requirements and the total proteome. The labels "A", "B", and "C" represent distinct phases based on changes in simulated fluxes through pathways with increasing glucose uptake flux: "A" corresponds to increase in both pathway 1 and 3, "B" decrease in pathway 3, and "C" decrease in pathway 1 while increase in pathway 2. The bottom three plots show sensitivities of glucose uptake, total proteome, and arginine uptake to the ATP production. The simulation data are reused as transparent lines in Appendix Fig S4.

the range of glucose uptake rates. We found that glucose and arginine uptake show high sensitivity scores when the upper limit of proteome is not reached, independently of variations in the

parameters (Appendix Fig S4). This demonstrates the robustness of the first finding. We further confirmed that the arginine shift can be interpreted by proteome constraints: The decline of arginine uptake once the total proteome is constrained was observed in almost all cases in which parameters were adjusted (Appendix Fig S4). Notably, the arginine shift delayed, but still in the proteome-limited regime, when the protein cost of mixed acid fermentation pathway increased (Appendix Fig S4A).

We conclude that the predicted changes in behavior, and the governing constraints that explain them, are robust to individual parameters, as well as to more global parameters that indicate the overall pathway costs.

**Experimental validations confirm glucose and arginine uptake as evolutionary changes for fitness improvement**

Given that the model predictions are based on optimization of growth, we conjectured that the reduced costs that represent active growth-limiting constraints, i.e., glucose and arginine uptake, should provide targets for fitness improvements. For the simulations, we used previously experimentally determined constraints on the uptake of amino acids (Goel *et al*, 2015), and we predicted that these could be improved under glucose-limited chemostat conditions according to our simulations—with arginine as the prime target (Fig 2). We therefore compared wild-type *L. lactis* MG1363 with a mutant strain (designated 445C1) that was selected during long-term cultivation in glucose-limited chemostat conditions at D = 0.5/h and that harbors a point mutation in the global carbon catabolite repression regulator (CcpA) (Price *et al*, 2019). This CcpA mutant shows a twofold increase in mixed acid fermentation at the endpoint of the laboratory evolution experiment, while fermentation toward lactate is decreased (Price *et al*, 2019). This change in metabolism is consistent with the prediction that the total proteome constraint is not yet active at 0.5/h: An active total proteome constraint would provide a driving force toward lactate formation as it has a higher protein efficiency than mixed acid fermentation (Fig 3B).

To further validate predictions, we assessed protein allocation and amino acid metabolism of wild-type and CcpA mutant, not studied before. We therefore re-cultivated both strains in glucose-limited chemostats at D = 0.5/h and compared them at the proteome and metabolic level (Fig 4A). Principal component analysis of the proteomics data confirmed reproducibility (Appendix Fig S5). Apparent catabolic and total carbon balances were between 89 and 100% (Dataset EV3). We also found increased mixed acid and decreased lactic acid fermentation for the CcpA mutant compared with wild type (Appendix Fig S6), in agreement with the published data (Price *et al*, 2019).

The proteomics data showed that arginine catabolism, including its uptake system is significantly upregulated in the CcpA mutant compared with the wild type (Fig 4B, Appendix Fig S7). Consequently, we found an increased arginine uptake flux in the CcpA mutant (Fig 4C). Moreover, we found that no residual arginine was detected anymore in its supernatant (Fig 4D) while other amino acids could be still detected (Appendix Fig S8), even though some had nonzero-reduced costs as well (Fig 2B), likely because metabolism of these amino acids is not under control of CcpA (Zomer *et al*, 2007). Taken together, arginine could be the most effective amino

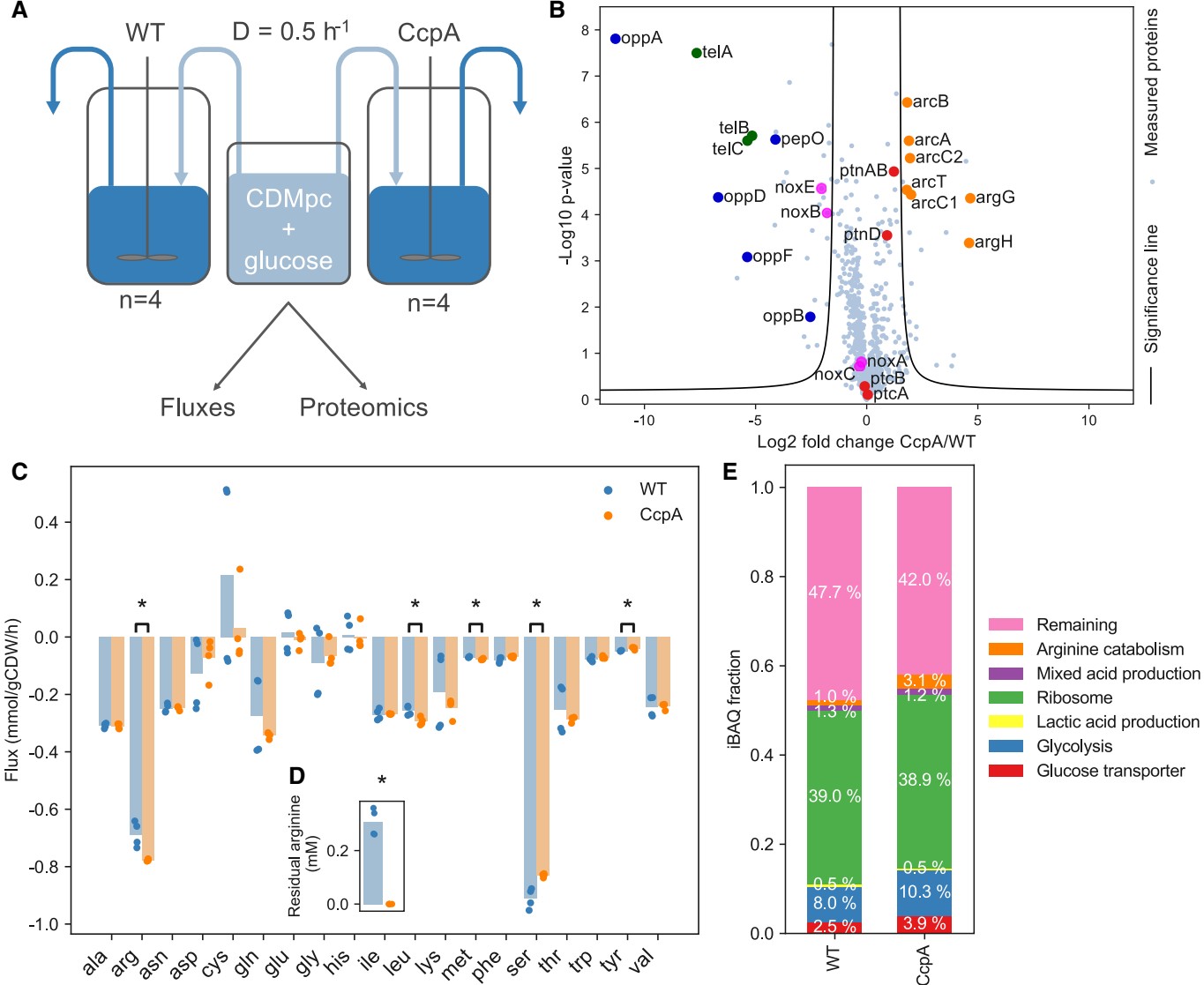

**Figure 4. Proteome of the CcpA mutant, which has a higher fitness than wild-type *L. lactis*, is changed in the direction of optimality.**

A   Overview of experiments for comparing wild-type *L. lactis* MG1363 and CcpA-mutant 445C1 in glucose-limited chemostats at D = 0.5/h. Samples were taken for dry weight measurements, external metabolite and amino acid analysis, and proteome measurements.

B   Protein fold changes of the CcpA mutant over wild type. Log2 fold changes were calculated from LFQ values. *P* values were calculated based on 2-sided two-sample *t*-tests with a FDR threshold of 0.05 and $S_0$ equal to 1 (Tusher *et al*, 2001). Proteins outside the significance lines (FDR = 0.05, $S_0$ = 1) are significantly changed in expression. Example sets of proteins are highlighted and are labeled with their gene names.

C, D   (C) Amino acid fluxes (mmol/gCDW/h) and (D) Residual concentration (mM) of arginine. The arginine uptake flux is increased to such extent that the residual arginine concentration became undetectable in the CcpA mutant. Data points represent the individual chemostat fluxes and concentrations; bars are average values (*n* = 4). Statistically significant differences between amino acid fluxes were calculated with two-sided *t*-tests and marked with * (*P* < 0.05).

E   Protein fractions. Protein fractions were obtained by normalizing iBAQ values by the total sum of all values. The obtained protein fractions were then averaged over the biological replicates (*n* = 4) for visualization. The fractions glucose transport, glycolysis, arginine pathway, and remaining fraction are significantly different between the two strains (*P* < 0.001, calculated with two-sided *t*-tests).

acid on optimizing fitness under the chosen conditions as predicted by our model.

We further found a significantly increased protein fraction of glucose transporters in the CcpA mutant (*P* < 0.001) (Fig 4E), confirming that glucose uptake capacity was growth limiting in the wild type. Even though the total proteome is not limiting at D = 0.5/h,

we did find changes in cytosolic protein fractions, notably the glycolysis fraction (Fig 4E), while other proteins, not used by pcLactis under glucose limitation, were downregulated, e.g., peptide digestion and NADH oxidases (Fig 4B, Appendix Fig S7). These may be part of the global catabolite repression effect of the CcpA mutation without large fitness impact. Alternatively, upregulation of

glycolysis may relieve inhibition on glucose transport, such as the negative impact of fructose 1,6-bisphosphate on the PTS system, via HPr (Deutscher *et al*, 2006).

We did not find a considerable change in the total protein membrane fraction (Appendix Table S1), suggesting that the membrane protein occupancy is independent of the tested conditions and, given the need to transport many nutrients, possibly maximally occupied. If fully occupied, an increase in one protein would go at the cost of another protein. In the CcpA mutant, we found an overrepresentation of significantly changed membrane proteins ($P = 0.030$ compared with random distribution of significantly changed proteins over membrane and cytosolic protein fraction, Appendix Table S1), among which downregulation of unused transport systems such as for peptides (Opp operon in Fig 4B). The idea that the membrane is fully occupied with proteins would also explain why amino acid uptake is not higher in wild type. It is therefore anticipated to impose a constraint on total membrane proteins in future simulations when transporter kinetics are available.

Altogether, the experimental data support model predictions and demonstrate that it is possible to integrate flux data with proteome-constrained models to identify growth-limiting factors through sensitivity analysis. These sensitivities are predictive for evolutionary change, providing deeper understanding of the driving forces that shape regulatory strategies.

## Discussion

It was previously shown that the metabolic shift from mixed acid to lactic acid fermentation in *L. lactis* was not accompanied by changes in protein expression of the corresponding pathways. Another shift, however, i.e., the repression of arginine catabolism at high growth rates, did coincide with the changes in protein levels (Goel *et al*, 2015). We developed a proteome-constrained model that can explain the inhibition of arginine catabolism at high growth rates as a result of a proteome limitation (Fig 2A), which is caused by a relatively lower protein efficiency of arginine catabolism compared with other ATP-producing pathways (Fig 3B). Therefore, our study demonstrates that also *L. lactis* appears to abide to the microbial growth laws of cellular resource allocation (Scott *et al*, 2010) after all. For this microorganism, that grows in nutrient-rich environments, however, constraints involve amino acid catabolism.

To cope with the many nutrients that *L. lactis* takes up in a growth medium with all amino acids present, we used reduced cost analysis, a well-known sensitivity analysis in constrain-based modeling (Teusink *et al*, 2006). We applied it to a proteome-constrained model and thus could balance the growth rate benefit of each metabolic activity with its associated protein costs. In this way, we identified the shifts in sensitivities in amino acid catabolism, most notably in arginine metabolism, that explained its repression at high growth rate.

To further test the model, we conjectured that these sensitivities should also point toward targets of growth improvements that selection should favor. We took advantage of a previous study where a CcpA mutant of *L. lactis* was selected and shown to have higher fitness under glucose-limited chemostats (Price *et al*, 2019). Here, we cultivated wild type and the evolved mutant again in glucose-limited chemostats at D = 0.5/h, where the model predicted that glucose and arginine uptake were dominant at limiting growth rates, and collected proteomics and flux data. We found that the evolved strain indeed upregulated protein levels of glucose and arginine catabolism. Therefore, we see the consistency between model predictions and experiments.

In spite of such consistencies, we also found discrepancies between experiments and model predictions. While pcLactis was able to predict a metabolic switch between fermentation pathways, experimentally the metabolic switch occurs at a much lower growth rate (Fig 1C). In the model, by design, the switch is accompanied by changes in protein levels, while in the experiments protein levels remain constant during the switch (Appendix Fig S2). This points to metabolic regulation of the switch, in line with the conclusion that protein costs cannot explain the switch (Goel *et al*, 2015). An explanation of the metabolic switch thus requires details on enzyme kinetics that are beyond the current stoichiometric model. Previously, it has been argued that the intracellular $NADH/NAD^+$ ratio causes the metabolic switch: The $NADH/NAD^+$ ratio increases with increasing glycolytic flux, which activates lactate dehydrogenase (LDH) and inhibits glyceraldehyde-3-phosphate dehydrogenase (GAPDH), which has NADH as a product (Garrigues *et al*, 1997; Even *et al*, 1999). This suggests that in order to sustain a high glycolytic flux, *L. lactis* needs rapid reversion of NADH into $NAD^+$, and this would be better achieved by LDH than mixed acid formation. This may explain why glyceraldehyde-3-phosphate and dihydroxyacetone phosphate have an inhibitory effect on pyruvate formate lyase (PFL) (Garrigues *et al*, 1997; Melchiorsen *et al*, 2001), the first enzyme of the mixed acid branch: The accumulation of these metabolites may indicate a bottleneck in GAPDH activity and the need to switch to LDH to lower the $NADH/NAD^+$ ratio. An alternative model proposed that differential protein costs of the two branches, caused by ATP inhibition, could cause a metabolic switch to occur while protein levels stay constant (de Groot *et al*, 2019). Current explanations for this metabolic switch thus remain speculation and require more detailed kinetic studies.

In addition, a few discrepancies come from amino acid predictions. Besides arginine, pcLactis also indicated other amino acids having a switch in the uptake rate in the proteome-limited regime, which were not found to be downregulated in the experiments. Most prominent are glycine, threonine, and serine (Fig 2). The predicted switch in glycine uptake is caused by a model artifact that consists of a (flux-limited) folate cycle that converts $NADP^+$ into NADPH at lower growth rates and switches direction at higher growth rates. Fluxes are small, however, and thus the impact is limited (Fig 2). Threonine can be converted by threonine aldolase (GlyA) to glycine and acetaldehyde (Aller *et al*, 2015), and the reduction in acetaldehyde to ethanol allows more acetate and hence ATP to be produced from pyruvate, explaining the positive effect of its catabolism on growth. The experimental data at D = 0.6/h are inconclusive (Goel *et al*, 2015), however, and the reduced costs are lower than that of arginine and serine.

Serine catabolism also has a beneficial effect on growth: Serine is converted to pyruvate via a deaminase reaction catalyzed by SdaA

and SdaB, and subsequent conversion to acetate would yield 1 ATP. The model indeed predicts maximal uptake of serine, consistent with the data. The much higher critical growth rate for the decrease in serine uptake (and catabolism) than for arginine catabolism can be explained by the lower protein costs for serine than for arginine (Appendix Table S2). Based on the reduced cost analysis, serine was predicted to have a significant impact on growth (Fig 2B), but the CcpA mutant decreased serine uptake (Fig 4C). This is probably the result of the relatively short evolutionary time, which allowed only the mutation with the highest selection coefficient to be selected, which was in CcpA. The higher ATP yield of arginine, but its lower proteome efficiency, explains why arginine is under glucose repression and serine is not. We suspect that the decrease in serine flux is the result of product inhibition by the CcpA-induced changes in mixed acid fermentation, but we can only speculate.

In conclusion, we develop a fine-grained proteome-constrained model of *L. lactis* and show that sensitivity analysis with the model enables identification of testable targets for improving microbial fitness in complex nutrient environments. We find that, besides glucose, arginine uptake is a growth-limiting factor of *L. lactis* up until a critical growth rate where protein costs start to outweigh its metabolic benefit. Arginine can be considered as an alternative ATP source, and its regulation is also analogous to the responses of microbes upon mixtures of carbon substrates, including catabolite repression by a preferred carbon source—glucose in our case. Simultaneous utilization is only observed for mixtures of substrates present at low concentrations (Lendenmann *et al*, 1996; Okano *et al*, 2019), or when they enter metabolism at different sites (more precisely, their co-consumption is part of one elementary flux mode (de Groot *et al*, 2019)) and metabolic benefits outweigh protein costs (Hermsen *et al*, 2015). We believe that our findings, approaches, and the proteome-constrained model can help to interpret and explain complex co-utilization phenotypes in microbes that grow in complex nutrient environments.

# Materials and Methods

### Reagents and Tools table

| Reagent/Resource | Reference or source | Identifier or catalog number |
|---|---|---|
| **Experimental models** | | |
| MG1363 (*Lactococcus lactis ssp. cremoris*) | Gasson (1983), Wegmann *et al* (2007) | |
| 445C1 (derivative of MG1363) | Price *et al* (2019) | |
| **Chemicals, enzymes and other reagents** | | |
| CDMpc with 25 mM glucose | Price *et al* (2019) | |
| Norvaline | Sigma-Aldrich | N7502-25G |
| Phthaldialdehyde | Sigma-Aldrich | P1378-25G |
| Trypsin | Roche | 11047841001 |
| **Software** | | |
| MetaDraft v0.7.2 | http://doi.org/10.5281/zenodo.2398336 | |
| CBMPy v0.7.25 | https://doi.org/10.5281/zenodo.3485023 | |
| Python 2.7 | https://www.python.org/ | |
| Cobratoolbox | https://github.com/opencobra/cobratoolbox/ | |
| Matlab | https://mathworks.com/ | |
| GECKO toolbox | https://github.com/SysBioChalmers/GECKO | |
| Soplex v4.0.0 | https://soplex.zib.de/ | |
| MaxQuant v1.6.6.0 | Cox & Mann (2008) | |
| Perseus v1.6.2.1 | Tyanova *et al* (2016) | |
| **Other** | | |
| HPLC | Shimadzu system | |
| Agilent Zebra Eclipse plus solvent saver 3.0 × 150 × 3.5 column | Agilent | |
| Needle sonicator | MPE | |
| Pall 3K omega filter | Sigma-Aldrich, The Netherlands | |
| AppliSens pH sensor | Applikon | |
| LTQ-OrbitrapXL | Thermo Scientific | |
| EASY-nLC1000 | Thermo Scientific | |

## Methods and Protocols

### Proteome-constrained model construction

The detailed construction procedure is described in the Appendix. Firstly, we updated the existing genome-scale metabolic model of *L. lactis* MG1363 (Verouden *et al*, 2009; Flahaut *et al*, 2013) in terms of transport reactions and gene-protein-reaction (GPR) associations by using MetaDraft (http://doi.org/10.5281/zenodo.2398336), a tool that reconstructs genome-scale metabolic networks based on previous manually curated ones by homology between genes, and subsequently manual curation. For non-spontaneous reactions with missing GPRs, we assigned a "dummy" protein as their catalysts to eliminate potential bias toward using them in simulations. Then, we split reactions with isozymes into multiple reactions that each is catalyzed by one isozyme, and reversible reactions into forward and reverse direction. In addition to updating and reformulating metabolic reactions, we formulated reactions for transcription, stable RNA cleavage, mRNA degradation, tRNA modification, rRNA modification, tRNA charging, ribosomal assembly, translation, protein maturation, protein assembly, enzyme formation, and protein degradation. Additionally, we formulated dilution reactions for RNA and enzymes to represent their dilution to daughter cells during cell division. Lastly, we modified the biomass equation of the metabolic model, i.e., removed protein and RNA from the equation as they were represented by dilution reactions, and added an unmodeled protein to account for all the other proteins that are not synthesized by the model. The genome-scale metabolic model was updated using CBMPy (https://doi.org/10.5281/zenodo.3485023), while pcLactis was constructed using the COBRA toolbox (Heirendt *et al*, 2019).

### Constraints and kinetics parameters

In addition to classical constraints of GEMs, e.g., mass balance and bounds on reaction rates, two protein constraints are imposed including a fixed bound on the total modeled proteome and an upper bound on the abundance of glucose transporter (Appendix). The fraction of modeled proteome was estimated according to the PaxDb database (Wang *et al*, 2015), where abundance of each protein is collected. The data are however available only for *L. lactis* IL1403, and thus, we performed BLASTp for mapping protein IDs between *L. lactis* IL1403 and MG1363. As a result, we obtained a list of proteins in *L. lactis* MG1363 with available abundance data (Dataset EV2). It should be noted that we filtered out the proteins of blocked reactions that would never carry fluxes in the model.

In addition, pcLactis accounts for coupling constraints that relate enzymes and machineries to their catalytic functions. In pcLactis, such constraints are imposed for coupling enzymes to metabolic reactions, RNA polymerase to transcription reactions, ribosomes to translation reactions, and so on. All the coupling constraints are detailed in the Appendix. In order to determine coefficients in the coupling constraints, we automatically retrieved turnover rates of metabolic enzymes from the BRENDA database (Jeske *et al*, 2019) using the GECKO toolbox (Sánchez *et al*, 2017) but also manually adjusted some according to literatures. In addition, we estimated catalytic rates of gene expression machineries, including ribosome, RNA polymerase, mRNA, and tRNA, in the same way as done for ME-Model of *E. coli* (O'Brien *et al*, 2013) based on reported data (Beresford & Condon, 1993; Novák & Loubiere, 2000) for *L. lactis*. Detailed description can be found in the Appendix.

### Simulations with pcLactis for anaerobic glucose-limited conditions

Since growth rate is integrated into coupling constraints in linear programming, it should be used as input for simulations. Therefore, we used a binary search workflow to obtain the minimal extracellular glucose concentration that leads to a feasible solution for any given growth rate to simulate glucose-limited conditions. This approach is based on the Michaelis–Menten equation with a fixed upper limit on the concentration of the glucose transporter:

$$q_S = k_{cat}[E]\sigma = k_{cat}[E]\frac{S}{K_M + S}, [E] \leq upper\ bound$$

in which $q_S$ is the glucose uptake rate, $k_{cat}$ is the turnover rate of the glucose transporter adopted from *E. coli* (Szenk *et al*, 2017), $[E]$ is the concentration of the glucose transporter, $\sigma$ is the saturation of the glucose transporter, $S$ is the extracellular glucose concentration and $K_M$ is the Michaelis constant obtained from the study (Price *et al*, 2019). The upper bound on the concentration of the glucose transporter was estimated using pcLactis, i.e., the minimal concentration that gives a feasible solution for the maximal growth rate. According to the Michaelis–Menten equation, searching for the minimal extracellular glucose concentration is equal to searching for the lowest saturation $\sigma$, which is equivalent to minimizing the glucose uptake rate when the upper bound of the concentration of the glucose transporter is hit.

The binary search solved successive individual linear programs, which maximizes the production of the dummy protein subject to stoichiometric and coupling constraints. Lower and upper bounds of some reactions were also constrained. Notably, amino acid exchange fluxes were constrained by growth rate-dependent upper bounds based on experimental data (Goel *et al*, 2015). In addition, biomass dilution reaction, which represents the dilution of other biomass components than modeled protein and RNA, was fixed according to the growth rate. Due to the anaerobic condition, we blocked two reactions, i.e., oxygen exchange reaction and pyruvate oxidase reaction. In addition, we blocked one alcohol dehydrogenase reaction catalyzed by the isozyme llmg_0955, and two glucose transport reactions due to low protein levels (Goel *et al*, 2015). Simulations were solved using Soplex 4.0.0 (https://soplex.zib.de/).

### Small model and simulations

The small model was extracted from pcLactis, which contains three pathways including glycolysis pathway with mixed acid fermentation, glycolysis pathway with lactic acid fermentation, and arginine catabolism.

We estimated ATP yield, protein cost, and protein efficiency for each pathway. Firstly, we identified the flux distribution of each pathway by solving a linear program with the metabolic part of pcLactis. For each simulation, the upper bound on ATP maintenance reaction was set free in order to account for ATP consumption. For mixed acid fermentation, the linear program is to maximize ATP maintenance reaction subject to a fixed glucose uptake rate of 1 mmol/gCDW/h. For lactate acid fermentation, the linear program is to maximize lactate production subject to a fixed glucose uptake rate of 1 mmol/gCDW/h and ATP maintenance reaction rate of 2 mmol/gCDW/h. For arginine catabolism, the linear program is to maximize ATP maintenance reaction subject to a fixed arginine uptake rate of 1 mmol/gCDW/h. Accordingly, we

calculated ATP yield for each pathway, i.e., the flux of ATP maintenance reaction over uptake rate of substrate. Secondly, we estimated the protein cost for each pathway based on the flux distribution, which is the protein cost of each reaction in the pathway times the corresponding flux value. The protein cost of a reaction is molecular weight of the corresponding enzyme over its turnover rate (Chen & Nielsen, 2019) and therefore can be extracted from pcLactis. Lastly, the protein efficiency of each pathway is ATP yield over protein cost.

With the parameters, we generated a linear program of fluxes through the three pathways, which is to maximize ATP production flux subject to constraints on uptake fluxes of substrates, i.e., glucose and arginine, and total proteome. We performed the simulations for a wide range of glucose uptake fluxes.

### Scaled reduced cost analysis

We performed the scaled reduced cost analysis (Teusink *et al*, 2006) for uptake rate of each amino acid on growth rate with pcLactis. The scaled reduced cost $R_i$ of the growth rate $\mu_i$ with respect to the uptake rate of an amino acid $q_i$ is calculated as: $R_i = (\Delta\mu/\Delta q)(q_i/\mu_i)$. We imposed a small increase in the uptake rate of each amino acid $\Delta q = 0.01$ to investigate the change in growth rate $\Delta\mu$.

### Sensitivity analysis

We performed sensitivity analysis for two constraints with pcLactis, i.e., the glucose transporter and modeled proteome, on growth rate. The sensitivity score $S_i$ of the growth rate $\mu_i$ with respect to a given constraint $c_i$ is calculated as: $S_i = (\Delta\mu/\Delta c)(c_i/\mu_i)$. With pcLactis, we imposed a small increase $\Delta c = 0.01$ in the glucose transporter and modeled proteome to investigate the change in growth rate $\Delta\mu$ at different conditions. We also performed sensitivity analysis with the small model for glucose uptake, proteome allocated to the small model and arginine uptake using the small increase $\Delta c = 0.01$ to investigate the change in ATP production rate $q_{ATP}$, i.e., the sensitivity score $S_i$ is calculated as: $S_i = (\Delta q_{ATP}/\Delta c)(c_i/q_{ATP})$.

### Strains and media

*Lactococcus lactis* ssp. *cremoris* MG1363 is a plasmid cured derivative of strain NCDO712 (Gasson, 1983; Wegmann *et al*, 2007). Strain 445C1 is a derivative of MG1363 isolated after prolonged cultivation in a chemostat (Price *et al*, 2019). Cultivation of strains was performed in chemically defined medium for prolonged cultivation (CDMpc) (Price *et al*, 2019), supplemented with 25 mM glucose as limiting carbon source at 30°C.

### Chemostat cultivation

Proteome, metabolite, and amino acid samples were obtained from four steady-state chemostats. Chemostat cultivation took place in 300 ml bioreactors with 270 ml working volume, under continuous stirring (using magnetic stirrers), while standing in 30°C water baths. The headspace was continuously flushed with 5% $CO_2$ and 95% $N_2$. pH was controlled at 6.5 using 2.5 M NaOH. The pH probe was calibrated before and after the pre-culture in the reactor. CDMpc with 25 mM glucose was added at a rate of 2.25 ml/min. Superfluous liquid was continuously removed from the top of the reactor to maintain a volume of 270 ml. This results in a dilution rate of 0.5/h. The flow rate of each medium pump was calibrated right before inoculating the reactor and checked again at

the end of the experiment. The exact volume of liquid in each reactor was determined at the end of the experiment by weighing the reactor liquid.

To start up a chemostat culture, 5 ml CDMpc with 25 mM glucose was inoculated with glycerol stocks of the respective strains, taken from a −80°C. This was directly added to the reactor containing 265 ml fresh 25 mM glucose CDMpc. The reactor was operated in batch mode (no medium addition, pH regulated, effluent pump on, 30°C, continuous stirring, headspace sparged) for 24 h, allowing the cells to reach the stationary phase. This was verified by observing that the pH remained constant without the addition of 2 M NaOH. After 24 h, the medium pump was switched on. The chemostat was operated for 8 volume changes before samples were taken. Samples were taken from the effluent of the reactor. Retention time in the effluent tube was < 2 min. Sample tubes were kept on ice while sample was being collected. For each strain, four replicate chemostats were cultivated.

### Sampling procedures and measurements

For extracellular metabolite concentration measurements (for both reactor broth and sterile medium), 2 ml samples were centrifuged at 27,237 *g* for 3 min at 4°C, and the supernatant was filtered through a 0.22 µm polyether sulfone filter (VWR international) and stored at −20°C until further analysis. Extracellular concentrations of lactate, acetate, formate, ethanol, and glucose were determined by high-performance liquid chromatography (HPLC) as described previously (Goel *et al*, 2012a).

Extracellular concentrations of amino acids were determined by HPLC on a Shimadzu system with: LC-20AD pumps, DGU-14A degasser, SIL-10ADvp autosampler, CTO-10ASvp oven, SCL-10Avp system controller, and RF-10AXL fluorescence detector. Separation occurred on an Agilent Zebra Eclipse plus solvent saver $3.0 \times 150 \times 3.5$ column over a period of 38 min per sample, with an isocratic flow of two eluents both at a flow rate of 0.64 ml/min. Eluent 1 had composition: 0.142% w/v $NaHPO_4$, 0.381% w/v $Na_2B_4O_7 \cdot 10H_2O$, 0.0325% w/v sodium azide in deionized water (DI water); and eluent 2 had composition: 45% v/v methanol, 45% v/v acetonitrile in DI water. Samples were prepared by mixing 25 µl sample with 875 µl DI water, 25 µl Borate buffer (0.6% w/v boric acid, 0.4% w/v NaOH, pH 10.2), and 25 µl 1 mM Norvaline as internal standard. For amino acid derivation, 3 µl phthaldialdehyde (Sigma-Aldrich), i.e., OPA, was automatically added 3 min before sample injection (5 µl) into the column. Concentrations were determined by comparison of sample peak areas to those of a calibration curve identically run with representative amino acid concentrations.

Proteome samples were collected in low protein binding tubes, centrifuged at 27,237 *g* for 3 min at 4°C. The supernatant was discarded, and the pellet was frozen in liquid $N_2$ prior to storage at −20°C until further analysis.

For dry weight measurements, 10 ml broth was collected and filtered over a pre-dried (3 days, 60°C) pre-weighed 0.2 µm cellulose nitrate filter (Whatman GmbH). The filter was then washed once with 10 ml demineralized water and dried (3 days, 60°C), before weighing again.

Fluxes ($q_i$ in mmol/gCDW/h) were calculated as: $q_i = D \cdot (C_{i,supernatant} - C_{i,medium})/X_{biomass}$, where D is the dilution rate (/h), C the concentration of compound $i$ (mM) and $X_{biomass}$ the biomass concentration (g/l).

### Proteome measurements

Proteins were isolated using the FASP method as described previously (Wiśniewski *et al*, 2009). In short, cell pellets were suspended in 100 mM TRIS pH 8 to a concentration of 7.5E8 cells/100 μl and lysed using a needle sonicator (MPE). Proteins obtained from 4.5E8 lysed cells were reduced with 15 mM dithiothreitol for 30 min at 45°C and subsequently alkylated in 20 mM acrylamide for 10 min at room temperature under denaturing conditions (100 mM TRIS pH8 + 8 M Urea). The alkylated sample was transferred to an ethanol-washed Pall 3K omega filter (Sigma-Aldrich, The Netherlands) and centrifuged for 36 min at 13,523 *g*. The filters were washed with 130 μl 50 mM ammonium bicarbonate and centrifuged again before overnight digestion with 100 μl 5 ng/μl trypsin (Roche) at room temperature. The digested peptides were eluted by centrifugation for 30 min, subsequent addition of 100 μl 1 ml/l HCOOH in water, and again centrifugation for 30 min. pH was adjusted to pH 3 using 10 v/v% trifluoroacetic acid.

nLC-MS/MS analysis was done with a Thermo EASY nLC1000 connected to a Thermo LTQ Orbitrap XL as described previously (Lu *et al*, 2011; Wendrich *et al*, 2017). Raw datafiles were analyzed using MaxQuant (version 1.6.1.0) and searched against the *L. lactis* MG1363 database (UniProt) and frequently observed contaminants. In addition to the standard settings, Trypsin/P with a maximum of two missed cleavages was set as the digestion mode, acrylamide modifications on the cysteines were set as a fixed modification, and methionine oxidation, protein N-terminal acetylation, and asparagine or glutamine deamidation were set as variable modifications. A false discovery rate of 1% at protein and peptide level was allowed, and the minimum required protein length was set at 7. At least two peptides were required for protein identification of which at least one peptide was required to be unique in the database. Identified proteins were quantified with MaxQuant's LFQ algorithm (Cox *et al*, 2014).

### Proteomics data analysis

Statistical analysis on the MaxQuant output was performed with Perseus version 1.6.2.1. Proteins were accepted when they were represented in at least three of the four biological replicates in at least one strain. For statistical analysis, log10 transformed LFQ values were used and zero values were replaced by taking random values from a normal distribution with mean (measured values per biological sample −1.8) and variation (0.3 * standard deviation of the measured values per biological replicate) to make calculations possible. We then performed a 2-sided two-sample *t*-tests using the log10 normalized LFQ intensity columns of CcpA mutant and wild type with a FDR threshold of 0.05 and $S_0$ equal to 1 (Tusher *et al*, 2001). Fold changes of CcpA mutant over wild type were calculated by dividing LFQ intensity columns of CcpA mutant by wild type. A principal component analysis of the LFQ data was used to assess the reproducibility of the data. Further analysis steps were done with Python. For fractions of protein groups, we used sums of intensity-based absolute quantitation (iBAQ) protein fractions, which were obtained by normalizing iBAQ values by the total sum of all values per biological replicate. Two-sided two-sample *t*-tests were performed on biological groups of proteins between the wild-type and CcpA-mutant strain. Furthermore, a two-sided Fisher exact test was performed on the number of significantly changed proteins and unchanged proteins in the membrane and rest of the cell. For this,

the protein fractions were first averaged over the biological replicates.

## Data availability

The datasets and computer code produced in this study are available in the following databases:

- Models and scripts: GitHub (https://github.com/SysBioChalmers/pcLactis).
- Mass spectrometry proteomics data: the ProteomeXchange Consortium via the PRIDE (Vizcaíno *et al*, 2016) partner repository with the dataset identifier PXD021956 (https://www.ebi.ac.uk/pride/archive/projects/PXD021956).

**Expanded View** for this article is available online.

## Acknowledgements

Y.C. and J.N. acknowledge funding from the European Union's Horizon 2020 research and innovation program under Grant Agreement No. 686070. Y.C. and J.N. also acknowledge funding from the Novo Nordisk Foundation (grant no. NNF10CC1016517). E.v.P.-K., B.v.O., S.B., H.B., D.M., and B.T. acknowledge funding from the Netherlands Organisation for Scientific Research (grant no. ALWTF.2015.4). S.D., H.B., D.M., and B.T. acknowledge funding from the Top-sector Agri&Food (grant no. AF-15503). The computations were partially enabled by resources provided by the Swedish National Infrastructure for Computing (SNIC) at HPC2N partially funded by the Swedish Research Council through grant agreement no. 2018-05973.

## Author contributions

Study conception: BT, HB, and JN; Modeling, simulations, and data analysis: YC and EvP-K; Experiments: BvO and SD; Data analysis: DM and SB; Manuscript writing: YC, EvP-K, and BT; Manuscript editing: All authors.

## Conflict of interest

The project is organized by and executed under the auspices of TiFN, a public–private partnership on precompetitive research in food and nutrition. The authors have declared that no competing interests exist in the writing of this publication. Funding for this research was obtained from Friesland Campina, CSK Food Enrichment, the Netherlands Organisation for Scientific Research and the Top-sector Agri&Food.

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
