## [Review Process File · Molecular Systems Biology]

Proteome constraints reveal targets for improving microbial fitness in nutrient-rich environments

Yu Chen, Eunice Pelt-KleinJan, Berdien Olst, Sieze Douwenga, Sjef Boeren, Herwig Bachmann, Douwe Molenaar, Jens Nielsen, and Bas Teusink

DOI: [10.15252/msb.202010093](https://doi.org/10.15252/msb.202010093)

Corresponding author(s): Bas Teusink (b.teusink@vu.nl)

Review Timeline:

Submission Date:	3rd Nov 20
Editorial Decision:	25th Nov 20
Revision Received:	9th Feb 21
Editorial Decision:	26th Feb 21
Revision Received:	26th Feb 21
Accepted:	1st Mar 21

Editor: Maria Polychronidou

Transaction Report:

Thank you again for submitting your work to Molecular Systems Biology. We have now heard back from the three referees who agreed to evaluate your study. Overall, the reviewers think that the presented model and findings seem relevant for the field. However, they raise a series of concerns, which we would ask you to address in a major revision.

Without repeating all the issues listed below, some of the more substantial concerns are the following:

- Reviewer #1 recommends performing a sensitivity analysis and examining how the chosen parameters and constraints affect the reported conclusions.
- Reviewer #1 also states that the main conclusions need to be articulated more clearly. During our cross-commenting process, in which the reviewers can make additional remarks based on each other's reports, reviewer #3 agreed with this recommendation and pointed out that the methodological and biological contributions of the work should be clearly described and discussed separately. Reviewer #3 also mentioned that if the intention of the study is to focus not only on the biological findings but also on the approach (= method to identify growth-limiting reactions), then more effort should be spent on benchmarking this approach.
- Reviewer #3 notes some discrepancies between experimental data and predictions, which will need to be addressed.
- Reviewer #3 also mentions that the prediction of the switch to combined glucose and proteome-limited growth (their point #2) needs to be better explained and supported.
- Several other points of the three reviewers refer to the need to further support and enhance the reported biological findings and clarify their novelty.

REFeree REPORTS

Reviewer #1:

It is unclear what the message of the paper is. Overall, I had difficulty really grasping the message and contribution of the paper. From the title alone, I didn't get it. Also the introduction doesn't help much as in part of the abstract, it seems that the authors contribute an "approach" (or "integrated analysis"). Also the last paragraph of the manuscript is very confusing and after having read it a couple of times, I still have difficulties stating (in my own words) of what we learned from this work. At some point, it seemed that the authors wondered whether the proteome constraints which others identified to hold under a single-nutrient condition would also hold true for multiple nutrient conditions. At some other points, it feels like the authors would like to propose their work as a method; at other points it comes across as a paper that the paper has a more evolutionary/fitness perspective.

Are the conclusions justified? Here, I am not certain for the following reason: The authors' model is not just a simple genome-scale metabolic network, but a model that also contains protein allocation constraints with enzyme kinetics and the processes of gene expression (transcription, RNA cleavage, mRNA degradation, etc.). This type of extension of a genome-scale metabolic network has been accomplished previously for some other organisms. This type of model requires a huge number of parameters (k_{deg} , k_{cat} , σ), many of which are completely unknown. Beyond, also constraints were imposed on upper and lower bounds of some reactions (see p. 7) and some reactions were also blocked (see p. 8). Given these huge number of required (uncertain) parameters and of constraints, I have no clue on in how far the global conclusions that the authors drew are dependent on any of these parameters and constraints. Could it be that the conclusions drawn are simply "artifacts" of parameters and constraints? While the authors did sensitivity analyses to identify which flux/constraint can increase growth rate/fitness, I think what would be required is an "analysis" on in how far the model parameters and constraints influence the overall conclusion. Without that, I cannot be convinced that any of the drawn conclusions is actually correct.

Another assumption of the model is that the authors use inactive enzyme to fill up the total proteome, at low growth rates. At a growth rate of 0.5 1/h, the authors find (in their simulations) that the fraction of inactive enzymes becomes zero. If I get everything right, then this would imply that from this growth rate onward, in a proteomics experiment, we should be able to identify *all* proteins (i.e. no presence of any unidentified proteins, as all known proteins are now modelled). If there are still unidentified proteins in a proteomics experiment above a growth rate of 0.5 1/h, this would mean that the model is not correct. Unfortunately, I cannot fully oversee what the consequences of this would be for the conclusions that are drawn here. Yet, it is these kinds of uncertainties that makes me doubt whether I should trust the model and the conclusions.

Overall, on the plus side, the manuscript is well-written, presents a nice model and an effort to integrate modeling effort with data. On the negative side: the model development itself reflects limited methodological innovation (as similar models were developed for other organisms), reflects limited (or at least very unclear) insights, which could even be artifacts of the model parameters and constraints.

Minor:

p. 2, lines 29-33: I feel that 2 sentences are contradicting each other

p2/l9: what are standard enzyme kinetics?

P2/l33: which model prediction?

P2/l47: You state that you lack info in expression and kinetics on amino acid transport systems, but is this also true for other enzymes?

P4/l29-33: Also glycine and threonine show a drop. What about those?

P5/l13: Why should this be so?

P5/l20: What is the point?

Fig4C: what about serine?

Fig1c: The model has lots of parameters (see above) and is being compared with limited experimental data. In the light of this, I find it puzzling that yet there is a rather significant deviation between the model prediction and the experimental observation. How come?

Fig. 4b: It is striking that many genes from the Opp operon are downregulated. Are these proteins part of the model as well? What does it mean that they are downregulated? Does the model predict this as well?

Reviewer #2:

Every now and then a manuscript comes along that addresses a relevant question in a convincing manner with clear and mature writing. In my view this is one such case. While resource allocation is an increasingly accepted factor for understanding microbial decisions, it is true that the evidence stems primarily from *E. coli* and yeast. By investigating resource allocation in an anaerobe, the authors convincingly demonstrate that the principle is more general. Over a large range of growth rates, cells were able to increase their flux by simply using enzymes of major pathways more extensively; ie they were more abundant than strictly necessary. Above a critical growth rate, however, this capacity was not sufficient anymore, and proteome allocation became a governing constraint, nicely capturing the observed changes in glucose and arginine catabolism. Consistency with the behavior of an evolved mutant, further provides evidence that the considered principles matter on an evolutionary time scale and can be used to correctly predict the course of evolution in a given environment.

I have only minor suggestions to further improve accessibility:

1. The nonchalant use of the term overflow metabolism might confuse many readers. Generally one considers aerobic fermentation as opposed to respiration as overflow, even if the term is clearly incorrect even then, if one is to think the resource allocation through. The reason for, for example, acetate formation, is not overflow (ie a limit in respiration) but a consequence of proteome allocation. Since the authors work with an anaerobe, it is very confusing to refer to lactate formation as overflow of mixed acid fermentation - even though it is technically comparable to acetate vs respiration. It would be best to avoid the term overflow altogether as it does not make sense in the context of proteome allocation. If the authors insist on using it, I would strongly recommend to at least define and explain it clearly for lactate formation right at the start.

2. Before going into the analysis of the evolution experiment (page 5), it would be very helpful to first explicitly formulate the predictions for the experiment so that a reader can appreciate consistency between theory and experiment.

Reviewer #3:

In the present work, Chen et al investigate the impact of proteome constraints on the metabolic program of *L. lactis* cultures growing in nutrient-rich environments. Towards this end, the authors develop a proteome-constrained FBA model of *L. lactis*, and integrate this model with previously published proteomics and physiology data from glucose-limited chemostats. Using this integrated model, the authors identify the uptake of glucose (expected given the nature of the glucose-limited chemostat experiments used here) and arginine as key constraints that limit *L. lactis* growth. The authors validate these predictions by showing that CcpA mutant strains (a mutation that was shown to consistently emerge in evolved glucose-limited chemostat cultures) do indeed increase arginine catabolism.

Overall, this manuscript is well written, and addresses an important and timely question. In particular, I appreciate the authors' efforts to utilize and re-interpret their previously published chemostat data. Moreover, the proteome-constrained FBA model (in my understanding, the first such model for *L. lactis*) will surely be useful to the readership of MSB. However, several key aspects of this work are currently unclear:

1. A key contribution of this work is the proteome-constrained FBA model of *L. lactis*. I appreciate the authors' efforts to derive testable predictions from this model (i.e. the importance of arginine catabolism for glucose-limited growth). However, the predicted fermentation product fluxes deviate quite substantially from experimental data (in the case of lactate, it is particularly hard to see the predicted threshold-linear behavior). Given this manuscript's focus on exactly these metabolic pathways, I would suggest a more in-depth discussion of this discrepancy (at least in a supplementary note).

2. The authors highlight the predictive power of their model by predicting the switch to combined glucose and proteome-limited growth (page 3, lines 29-33). However, it is currently difficult to appreciate the quality of this prediction, since the authors only show simulated proteome fractions in Figure S1. In my understanding, the authors' claim stems from their previous observation that there is genome-wide proteome reallocation (PMID 25828364) around the identified critical growth rate (between 0.5 and 0.6 h⁻¹). However, at least to me it was not apparent where exactly this statement was made in the cited paper. A direct comparison of measured and predicted proteome fractions in this manuscript would be more convincing.

3. In my understanding, the authors obtained the required enzyme *k*_{cat} values either from literature, or set unknown *k*_{cat} values to 100 s⁻¹ (median of literature *k*_{cat} values used here). How sensitive are the findings in this manuscript (e.g. regarding the predicted importance of arginine catabolism for growth) to uncertainties in reported *k*_{cat} values?

Additional comments:

1. A key insight of this work is that the onset of arginine catabolism (and NOT the transition from lactate to mixed acid fermentation, as previously speculated) coincides with the critical growth rate at which the total proteome constraint kicks in. Thus, in *L. lactis* arginine seems to play a similar role as alternative carbon sources play in other microbes (i.e. switch to co-utilization of preferred and alternative carbon sources below a critical growth rate in *E. coli*, see e.g. PMID 7894722 & PMID 31819215). This finding is very interesting, but currently relegated to the last paragraph of the manuscript. A more detailed discussion of this finding and its implications may benefit those readers who are more interested in the biology examined here rather than the computational approach.
2. In Figure 3B, the authors state that the ATP yield of arginine catabolism is 1.67. Based on the

authors' previous work (PMID 25828364), I would have naively expected an ATP yield of 1 (since it seems that only 1 ATP is produced in the carbamase kinase reaction, Figure 3A in PMID 25828364). Maybe the authors can clarify where exactly this additional ATP comes from?

3. I was surprised to see such a large difference in protein cost between mixed acid and lactic acid fermentation (Figure 3B), since these two pathways share many (glycolytic) enzymes. Similarly, I was surprised to see that arginine catabolism has a much higher protein cost compared to mixed acid/lactic acid fermentation, since the arginine metabolism proteome fraction seems to be substantially (1 versus 8%) smaller than the glycolysis proteome fraction. Are these differences in protein cost largely driven by enzymes with particularly low k_{cat} values?

4. Besides arginine, the authors identified serine as another amino acid whose catabolism is predicted to have a high impact on glucose-limited *L. lactis* growth (Figure 2B). Do the authors have an explanation for this observation, and/or any data to validate this prediction?

5. The authors predict that at growth rates above 0.5h^{-1} , most (if not all) enzymes in *L. lactis* are operating at maximal capacity. This prediction seems surprising given that (at least in *E. coli*) previous studies have shown that a substantial fraction of the "core" proteome is under-utilized (e.g. PMID 27351952). Maybe the authors can comment on this discrepancy in the text?

6. This manuscript relies heavily on data from the authors' previous publications (PMID 25828364, PMID 30630406), which of course is completely fine. However, I believe this manuscript would benefit from a clarifying paragraph, in which the authors outline which of the results presented here confirm their previous findings, and which results constitute novel insights/interpretations.

Reviewer #1:

It is unclear what the message of the paper is. Overall, I had difficulty really grasping the message and contribution of the paper. From the title alone, I didn't get it. Also the introduction doesn't help much as in part of the abstract, it seems that the authors contribute an "approach" (or "integrated analysis"). Also the last paragraph of the manuscript is very confusing and after having read it a couple of times, I still have difficulties stating (in my own words) of what we learned from this work. At some point, it seemed that the authors wondered whether the proteome constraints which others identified to hold under a single-nutrient condition would also hold true for multiple nutrient conditions. At some other points, it feels like the authors would like to propose their work as a method; at other points it comes across as a paper that the paper has a more evolutionary/fitness perspective.

The aim of our study was to investigate, with the aid of the developed proteome-constrained model and sensitivity analysis using the model, the behaviour of *Lactococcus lactis* under nutrient-rich conditions. As a consequence, we found 1) that the metabolic shift associated with arginine catabolism can be interpreted by proteome constraints, and 2) that glucose and arginine uptake limit growth when *L. lactis* is not proteome-constrained, which can be validated by our experimental data. We would not like to propose our work as a method as we think that our biological findings are more attractive although our experimental validations suggest the method used in the study, i.e., sensitivity analysis using pcLactis, has general applicability.

To improve the clearness of the manuscript, we have 1) added subtitles in the main text, and 2) substantially revised the last paragraph. Also, we have revised the remaining of the text to the best of our abilities to avoid the confusion.

Are the conclusions justified? Here, I am not certain for the following reason: The authors' model is not just a simple genome-scale metabolic network, but a model that also contains protein allocation constraints with enzyme kinetics and the processes of gene expression (transcription, RNA cleavage, mRNA degradation, etc.). This type of extension of a genome-scale metabolic network has been accomplished previously for some other organisms. This type of model requires a huge number of parameters (k_{deg} , k_{cat} , σ), many of which are completely unknown. Beyond, also constraints were imposed on upper and lower bounds of some reactions (see p. 7) and some reactions were also blocked (see p. 8). Given these huge number of required (uncertain) parameters and of constraints, I have no clue on in how far the global conclusions that the authors drew are dependent on any of these parameters and constraints. Could it be that the conclusions drawn are simply "artifacts" of parameters and constraints? While the authors did sensitivity analyses to identify which flux/constraint can increase growth rate/fitness, I think what would be required is an "analysis" on in how far the model parameters and constraints influence the overall conclusion. Without that, I cannot be convinced that any of the drawn conclusions is actually correct.

As suggested, we have investigated how far the model parameters and constraints influence our findings, and added this into the Results section with the subtitle "Predictions are robust to uncertainties of model parameters".

To this end, we examined parameters and constraints of both the genome-scale model pLactis and the small model. Regarding pLactis, we investigated in total 845 parameters and constraints, including turnover rates of metabolic enzymes, catalytic rates of gene expression machineries, degradation constants, upper limits on modelled proteome and glucose transporter, GAM (growth-associated ATP maintenance) and NGAM (non-growth-associated ATP maintenance). Although the reviewer also mentioned that we imposed bounds on some reactions and blocked some reactions, there are no uncertainties on these bounds as they were from measured data, which we already stated in the main text. Regarding the small model, we investigated five parameters and constraints, including protein costs of the pathways, and upper limits on proteome and arginine uptake. As can be seen in the main text, our findings are robust. We hope that with the addition of these analyses the reviewer will find our paper improved.

Another assumption of the model is that the authors use inactive enzyme to fill up the total proteome, at low growth rates. At a growth rate of 0.5 1/h, the authors find (in their simulations) that the fraction of inactive enzymes becomes zero. If I get everything right, then this would imply that from this growth rate onward, in a proteomics experiment, we should be able to identify *all* proteins (i.e. no presence of any unidentified proteins, as all known proteins are now modelled). If there are still unidentified proteins in a proteomics experiment above a growth rate of 0.5 1/h, this would mean that the model is not correct. Unfortunately, I cannot fully oversee what the consequences of this would be for the conclusions that are drawn here. Yet, it is these kinds of uncertainties that makes me doubt whether I should trust the model and the conclusions.

We apologize for not having clearly stated the term “inactive enzyme” in the manuscript. The inactive enzyme is not unidentified protein, but just assumed to account for the inactive fraction of undersaturated enzymes. Our model uses the maximal turnover rate of each enzyme across all simulated conditions and thus predicts a minimum proteome composition required for a given metabolic state, but this is not the case in reality where enzymes are usually undersaturated at low growth rates.

We have stated this clearer in the text as:

“It should be noted that the inactive enzyme does not represent any specific enzyme but just accounts for the sum of inactive fractions of all undersaturated enzymes. Therefore, the inactive enzyme can be seen as the excess capacity of the total modelled proteome, and the constraint of the total modelled proteome is reached if the inactive enzyme is zero.”

Overall, on the plus side, the manuscript is well-written, presents a nice model and an effort to integrate modeling effort with data. On the negative side: the model development itself reflects limited methodological innovation (as similar models were developed for other organisms), reflects limited (or at least very unclear) insights, which could even be artifacts of the model parameters and constraints.

Thank you for the comments and we hope that our biological insights are more clear in the revision.

Minor:

p. 2, lines 29-33: I feel that 2 sentences are contradicting each other

We have changed the sentences as such:

“It was previously shown that anaerobic “overflow” metabolism in the lactic acid bacterium Lactococcus lactis, i.e., the transition from energy-efficient mixed-acid fermentation to less energy-efficient lactic acid fermentation, is not accompanied by changes in associated protein levels (Goel et al, 2015), questioning the generality of the resource allocation paradigm. However, consistent changes in both gene expression and metabolic levels were observed in amino acid metabolism, prompting us to revisit the cellular economics of L. lactis.”

p2/I9: what are standard enzyme kinetics?

Should it be p3/I9? We have changed the sentence as such:

“We constrain each metabolic flux at the maximal rate of the associated enzyme, which is a function of the enzyme concentration and turnover rate; Thus, we compute minimal enzyme levels to sustain metabolic flux.”

P2/I33: which model prediction?

Should it be P3/I33? We have changed the sentences as such:

“In the model this means that all available proteome space is being actively used for metabolic function, and that under such conditions any flux change can only be brought about by changes in protein levels, not enzyme saturation. This conclusion is supported by the fact that most glycolytic enzymes reach the highest saturation above $D = 0.5 \text{ h}^{-1}$ (Goel et al, 2015), and that proteomics data showed genome-wide protein reallocation when growth rate increased beyond that point (Appendix Fig S2).”

P2/I47: You state that you lack info in expression and kinetics on amino acid transport systems, but is this also true for other enzymes?

Should it be P3/I47?

No, this is not true for other enzymes.

We collected turnover rates for as many as possible enzymes from the BRENDA database using the GECKO toolbox (PMID: 28779005) and then manually curated many key enzymes such as those in the small model based on literature. However, we still lack data of enzymes especially transporters. The reason why we put the sentence like this is to demonstrate that we are not able to impose membrane constraints currently, which would ideally constrain the uptake of amino acids without setting uptake rates a priori. In spite of the lack of kinetic data, we have demonstrated in the revision that our findings are robust to uncertainties of turnover rates.

P4/I29-33: Also glycine and threonine show a drop. What about those?

We have added our response as a part of Discussion section as such:

“The predicted switch in glycine uptake is caused by a model artefact that consists of a (flux-limited) folate cycle that converts NADP^+ into NADPH at lower growth rates, and switches direction at higher growth rates. Fluxes are small, however, and thus the impact limited (Fig

2). Threonine can be converted by threonine aldolase (GlyA) to glycine and acetaldehyde (Aller et al, 2015), and the reduction of acetaldehyde to ethanol allows more acetate and hence ATP to be produced from pyruvate, explaining the positive effect of its catabolism on growth. The experimental data at $D = 0.6 \text{ h}^{-1}$ is inconclusive (Goel et al, 2015), however, and the reduced costs are lower than that of arginine and serine.”

P5/I13: Why should this be so?

Our simulations indicate that there is a more optimal uptake pattern of amino acids (Fig 2B). Currently the uptake rates are constrained based on experimental data (Fig 2A). We show that in the evolved strain the uptake patterns are indeed changed for the condition it is evolved in, especially for arginine. We have adjusted the sentence as follows:

“For the simulations we used previously experimentally determined constraints on the uptake of amino acids (Goel et al, 2015), and we predicted that these could be improved under glucose-limited chemostat conditions according to our simulations – with arginine as the prime target (Fig 2).”

P5/I20: What is the point?

If the total proteome constraint is active, cells would prefer to use lactic acid fermentation rather than mixed acid fermentation to produce ATP due to the fact that lactic acid fermentation has a higher protein efficiency than mixed acid fermentation (Fig 3B). However, in practice we found that the evolved strain even secreted more mixed acids and less lactate, which therefore means that the total proteome constraint is not yet active under the dilution rate of 0.5 h^{-1} .

We have changed the sentence as such:

“This change in metabolism is consistent with the prediction that the total proteome constraint is not yet active at 0.5 h^{-1} : An active total proteome constraint would provide a driving force towards lactate formation as it has a higher protein efficiency than mixed acid fermentation (Fig 3B).”

Fig4C: what about serine?

We have added our response as a part of Discussion section as such:

“Serine catabolism also has a beneficial effect on growth: serine is converted to pyruvate via a deaminase reaction catalysed by SdaA and SdaB, and subsequent conversion to acetate would yield 1 ATP. The model indeed predicts maximal uptake of serine, consistent with the data. The much higher critical growth rate for the decrease in serine uptake (and catabolism) than for arginine catabolism can be explained by the lower protein costs for serine than for arginine (Appendix Table S2). Based on the reduced cost analysis, serine was predicted to have a significant impact on growth (Fig 2B), but the CcpA mutant decreased serine uptake (Fig 4C). This is probably the result of the relatively short evolutionary time, which allowed only the mutation with the highest selection coefficient to be selected, which was in CcpA. The higher ATP yield of arginine, but its lower proteome efficiency, explains why arginine is under glucose repression and serine is not. We suspect that the decrease in serine flux is the result of product inhibition by the CcpA-induced changes in mixed acid fermentation, but we can only speculate.”

Fig1c: The model has lots of parameters (see above) and is being compared with limited experimental data. In the light of this, I find it puzzling that yet there is a rather significant deviation between the model prediction and the experimental observation. How come?

It is as we expected that pLactis cannot predict very well the metabolic shift between mixed acid to lactic acid fermentation because it has already been found that protein costs cannot explain the shift. Our model predictions even support this finding given the significant deviation in terms of the shifting point. Therefore, we did not highlight such a shift in the manuscript. Instead, we did find that the metabolic shift in arginine uptake can be explained by protein costs, which is a novel finding of our study.

We have added our response to the inconsistency as a part of Discussion section. We have stated that the metabolic switch from mixed acid to lactic acid fermentation rather involves enzyme kinetics that are beyond the current stoichiometric model.

Fig. 4b: It is striking that many genes from the Opp operon are downregulated. Are these proteins part of the model as well? What does it mean that they are downregulated? Does the model predict this as well?

There are several Opp proteins in the model, which are assigned to catalyse transport of dipeptides. In our study there are no dipeptides in the medium. Thus, the model cannot predict the abundances or changes of these proteins when blocking exchange of dipeptides because the model predicts the optimality using linear programming. However, it seems that *L. lactis* still expresses Opp proteins even though there are no dipeptides in the environment. The expression of useless proteins could be a strategy for adaptation to fluctuating environments. The downregulation of the Opp proteins after evolution means that *L. lactis* becomes more optimal for the environment without dipeptides. This is just moving towards the model prediction, i.e., no Opp proteins should be produced in the condition without dipeptides.

Reviewer #2:

Every now and then a manuscript comes along that addresses a relevant question in a convincing manner with clear and mature writing. In my view this is one such case. While resource allocation is an increasingly accepted factor for understanding microbial decisions, it is true that the evidence stems primarily from *E. coli* and yeast. By investigating resource allocation in an anaerobe, the authors convincingly demonstrate that the principle is more general. Over a large range of growth rates, cells were able to increase their flux by simply using enzymes of major pathways more extensively; ie they were more abundant than strictly necessary. Above a critical growth rate, however, this capacity was not sufficient anymore, and proteome allocation became a governing constraint, nicely capturing the observed changes in glucose and arginine catabolism. Consistency with the behavior of an evolved mutant, further provides evidence that the considered principles matter on an evolutionary time scale and can be used to correctly predict the course of evolution in a given environment.

We thank the reviewer for the kind words.

I have only minor suggestions to further improve accessibility:

1. The nonchalant use of the term overflow metabolism might confuse many readers. Generally one considers aerobic fermentation as opposed to respiration as overflow, even if the term is clearly incorrect even then, if one is to think the resource allocation through. The reason for, for example, acetate formation, is not overflow (ie a limit in respiration) but a consequence of proteome allocation. Since the authors work with an anaerobe, it is very confusing to refer to lactate formation as overflow of mixed acid fermentation - even though it is technically comparable to acetate vs respiration. It would be best to avoid the term overflow altogether as it does not make sense in the context of proteome allocation. If the authors insist on using it, I would strongly recommend to at least define and explain it clearly for lactate formation right at the start.

Thank you for pointing this out. We do not think that the terminology of overflow metabolism is limited to aerobic fermentation vs. respiration, but we agree that it can be avoided in this paper. We make the link now once and talk about metabolic switches instead. Overflow metabolism can be defined as: the simultaneous metabolism of nutrients by an energy-efficient and a less energy-efficient pathway (PMID: 31758233). We therefore gave a definition along with the first occurrence of overflow metabolism in the context of *L. lactis* in the text:

"It was previously shown that anaerobic "overflow" metabolism in the lactic acid bacterium Lactococcus lactis, i.e., the transition from energy-efficient mixed-acid fermentation to less energy-efficient lactic acid fermentation"

2. Before going into the analysis of the evolution experiment (page 5), it would be very helpful to first explicitly formulate the predictions for the experiment so that a reader can appreciate consistency between theory and experiment.

We have changed the first sentence in the experiment section as such:

"Given that the model predictions are based on optimisation of growth, we conjectured that the reduced costs that represent active growth-limiting constraints, i.e., glucose and arginine uptake, should provide targets for fitness improvements."

Reviewer #3:

In the present work, Chen et al investigate the impact of proteome constraints on the metabolic program of *L. lactis* cultures growing in nutrient-rich environments. Towards this end, the authors develop a proteome-constrained FBA model of *L. lactis*, and integrate this model with previously published proteomics and physiology data from glucose-limited chemostats. Using this integrated model, the authors identify the uptake of glucose (expected given the nature of the glucose-limited chemostat experiments used here) and arginine as key constraints that limit *L. lactis* growth. The authors validate these predictions by showing that CcpA mutant strains (a mutation that was shown to consistently emerge in evolved glucose-limited chemostat cultures) do indeed increase arginine catabolism.

Overall, this manuscript is well written, and addresses an important and timely question. In particular, I appreciate the authors' efforts to utilize and re-interpret their previously

published chemostat data. Moreover, the proteome-constrained FBA model (in my understanding, the first such model for *L. lactis*) will surely be useful to the readership of MSB. However, several key aspects of this work are currently unclear:

We thank the reviewer for the kind summary.

1. A key contribution of this work is the proteome-constrained FBA model of *L. lactis*. I appreciate the authors' efforts to derive testable predictions from this model (i.e. the importance of arginine catabolism for glucose-limited growth). However, the predicted fermentation product fluxes deviate quite substantially from experimental data (in the case of lactate, it is particularly hard to see the predicted threshold-linear behavior). Given this manuscript's focus on exactly these metabolic pathways, I would suggest a more in-depth discussion of this discrepancy (at least in a supplementary note).

As mentioned in the comment, we would just like to highlight our novel finding on the arginine catabolism. However, we found the discrepancy in the metabolic shift between mixed acid to lactic acid fermentation. This is as we expected because it has already found that protein costs do not explain this shift. The discrepancy between experiments and our model predictions can even support this previous finding. We agree that it is necessary to discuss the discrepancy in the manuscript.

As suggested, we have added our response on the discrepancy as a part of Discussion section. We have stated that the metabolic switch from mixed acid to lactic acid fermentation rather involves enzyme kinetics that are beyond the current stoichiometric model.

2. The authors highlight the predictive power of their model by predicting the switch to combined glucose and proteome-limited growth (page 3, lines 29-33). However, it is currently difficult to appreciate the quality of this prediction, since the authors only show simulated proteome fractions in Figure S1. In my understanding, the authors' claim stems from their previous observation that there is genome-wide proteome reallocation (PMID 25828364) around the identified critical growth rate (between 0.5 and 0.6 h⁻¹). However, at least to me it was not apparent where exactly this statement was made in the cited paper. A direct comparison of measured and predicted proteome fractions in this manuscript would be more convincing.

Thank you for pointing this out. It is not possible to directly compare the predicted with measured protein levels as our model assumes maximal turnover rate for each enzyme across various growth rates and thus predicts the minimal proteome requirement, while the apparent turnover rates of many enzymes are in practice changed due to incomplete saturation with substrates and regulation. Predicting those effects would require detailed enzyme kinetics at genome scale that is not available. Direct comparison of predicted and measured proteins would therefore lead to discrepancies – the predictions show generally increasing protein levels of, for example, glycolytic enzymes with increasing growth rate (Appendix Fig S1); the measured data however show almost constant protein levels of individual glycolytic enzymes (Fig 4 in PMID: 25828364).

To evaluate the predictive power of the model, we decided to focus on inactive enzyme. The *in-silico* production of inactive enzyme indicates excess capacity of enzymes, which should

be allocated to a part of catalytically active enzymes in reality. Below the critical growth rate, the model produces inactive enzyme, meaning that it is not proteome constrained yet. According to the experimental data, we can also see that the proteome constraint is not active below the critical growth rate as many enzymes, e.g., glycolytic enzymes, do not change their protein levels but just increase apparent turnover rates to achieve increased fluxes as indicated by the V_{\max}/Flux column in Fig4 in (PMID: 25828364). Only above the critical growth rate we can see considerable changes in protein levels, meaning that the total proteome constraint becomes active and thus *L. lactis* has to optimally reallocate proteome resource. Accordingly, we demonstrate the predictive power of the model by showing that it finds the critical growth rate above which proteome constraint becomes active in practice.

As well-spotted by the reviewer, there is not a clear statement in (PMID: 25828364) on the genome-wide proteome reallocation above the critical growth rate, and we thank the reviewer for making this point. We have now added a supplementary figure (Appendix Fig S2) using the published proteomics data to show that protein levels of enzymes change considerably above the critical growth rate. In addition, we have changed the sentences as such:

“More importantly, pcLactis predicted that at a dilution rate (D) higher than 0.5 h^{-1} , the fraction of inactive enzyme becomes zero (Fig 1E). In the model this means that all available proteome space is being actively used for metabolic function, and that under such conditions any flux change can only be brought about by changes in protein levels, not enzyme saturation. This conclusion is supported by the fact that most glycolytic enzymes reach the highest saturation above $D = 0.5 \text{ h}^{-1}$ (Goel et al, 2015), and that proteomics data showed genome-wide protein reallocation when growth rate increased beyond that point (Appendix Fig S2).”

3. In my understanding, the authors obtained the required enzyme kcat values either from literature, or set unknown kcat values to 100 s^{-1} (median of literature kcat values used here). How sensitive are the findings in this manuscript (e.g. regarding the predicted importance of arginine catabolism for growth) to uncertainties in reported kcat values?

As suggested, we have investigated how sensitive the findings are to uncertainties in reported kcat values as well as other parameters, and added this into the Results section with the subtitle of “Predictions are robust to uncertainties of model parameters”. We found that our findings are independent on most of the parameters. Especially, when the total proteome constraint is not active, kcat values would almost have no impact on the simulations (Appendix Fig S3B).

Additional comments:

1. A key insight of this work is that the onset of arginine catabolism (and NOT the transition from lactate to mixed acid fermentation, as previously speculated) coincides with the critical growth rate at which the total proteome constraint kicks in. Thus, in *L. lactis* arginine seems to play a similar role as alternative carbon sources play in other microbes (i.e. switch to co-utilization of preferred and alternative carbon sources below a critical growth rate in *E. coli*, see e.g. PMID 7894722 & PMID 31819215). This finding is very interesting, but currently relegated to the last paragraph of the manuscript. A more detailed discussion of this finding

and its implications may benefit those readers who are more interested in the biology examined here rather than the computational approach.

Thank you for this very constructive comment! We have substantially revised the conclusion paragraph and have added the detailed discussion of co-utilization of multiple carbon sources.

2. In Figure 3B, the authors state that the ATP yield of arginine catabolism is 1.67. Based on the authors' previous work (PMID 25828364), I would have naively expected an ATP yield of 1 (since it seems that only 1 ATP is produced in the carbamate kinase reaction, Figure 3A in PMID 25828364). Maybe the authors can clarify where exactly this additional ATP comes from?

Thank you for pointing this out. We have clarified this in the legend of Figure 3 as such:
*“Note that the ATP yield of the arginine pathway is 1.67, which is due to the fact that besides one mole of ATP produced by the carbamate kinase reaction, by degrading one mole of arginine *L. lactis* does not have to export two moles of proton, which equivalently saves 0.67 mole of ATP that should have been consumed by ATP synthase to balance protons.”*

3. I was surprised to see such a large difference in protein cost between mixed acid and lactic acid fermentation (Figure 3B), since these two pathways share many (glycolytic) enzymes. Similarly, I was surprised to see that arginine catabolism has a much higher protein cost compared to mixed acid/lactic acid fermentation, since the arginine metabolism proteome fraction seems to be substantially (1 versus 8%) smaller than the glycolysis proteome fraction. Are these differences in protein cost largely driven by enzymes with particularly low *k_{cat}* values?

The large difference in protein cost between mixed and lactic acid fermentation is due to the fact that lactic acid fermentation has only one enzyme below the node of pyruvate, i.e., lactate dehydrogenase, while mixed acid fermentation has more enzymes and their turnover rates are all much lower than lactate dehydrogenase.

Regarding the arginine catabolism, we have to state that protein cost and proteome fraction are different. As clarified in the legend of Figure 3, protein cost is the protein required **for carrying one unit of flux**. The arginine proteome fraction is 1% while glycolysis is 8%, but the arginine uptake is over 0.7 mmol/gCDW/h while glucose uptake is over 15. The protein costs should be around 1%/0.7 versus 8%/15, i.e., 1.4% versus 0.53%, which is in line with our calculation.

4. Besides arginine, the authors identified serine as another amino acid whose catabolism is predicted to have a high impact on glucose-limited *L. lactis* growth (Figure 2B). Do the authors have an explanation for this observation, and/or any data to validate this prediction?

We have added our response as a part of Discussion section as such:

“Serine catabolism also has a beneficial effect on growth: serine is converted to pyruvate via a deaminase reaction catalysed by SdaA and SdaB, and subsequent conversion to acetate would yield 1 ATP. The model indeed predicts maximal uptake of serine, consistent with the data. The much higher critical growth rate for the decrease in serine uptake (and catabolism)

than for arginine catabolism can be explained by the lower protein costs for serine than for arginine (Appendix Table S2). Based on the reduced cost analysis, serine was predicted to have a significant impact on growth (Fig 2B), but the CcpA mutant decreased serine uptake (Fig 4C). This is probably the result of the relatively short evolutionary time, which allowed only the mutation with the highest selection coefficient to be selected, which was in CcpA. The higher ATP yield of arginine, but its lower proteome efficiency, explains why arginine is under glucose repression and serine is not. We suspect that the decrease in serine flux is the result of product inhibition by the CcpA-induced changes in mixed acid fermentation, but we can only speculate."

5. The authors predict that at growth rates above 0.5h^{-1} , most (if not all) enzymes in *L. lactis* are operating at maximal capacity. This prediction seems surprising given that (at least in *E. coli*) previous studies have shown that a substantial fraction of the "core" proteome is under-utilized (e.g. PMID 27351952). Maybe the authors can comment on this discrepancy in the text?

For modelling purpose, we assumed that at the maximal growth rate (unlimited growth under glucose-rich conditions) all enzymes are fully saturated/used, i.e., inactive enzyme is zero (Panel A in the figure below), which could not be correct in reality as at the highest growth rate the mean saturation of *E. coli* is less than 80% (Fig 1D of PMID 27351952). It is doable to assume that at the maximal growth all enzymes in pLactis are undersaturated by fixing the lower bound of the inactive enzyme to a none-zero value (Panel B in the figure below), then we can of course ensure that in all simulations the proteome is undersaturated. This is however unnecessary (but complicates explanations) as it will still lead to hitting a proteome constraint with all consequences on switches we show in the simulations (Appendix Fig S4).

However, we should emphasize here that pLactis correctly predicted the trend of the changes in enzyme saturation, i.e., saturation linearly increases with growth rate and then reaches the highest level above the critical growth rate where the total proteome is constrained. This is in line with most glycolytic enzymes in our previous study where we observed that the value of V_{\max}/Flux (reciprocal of saturation) decreases from $D = 0.15\text{h}^{-1}$ to $D = 0.5\text{h}^{-1}$ and keeps unchanged or slightly increased at $D = 0.6\text{h}^{-1}$ (Fig 4 of PMID: 25828364).

We prefer to not comment on this issue in the text to not make it more complicated. To avoid misunderstanding, we have revised the sentence as such:

"In the model this means that all available proteome space is being actively used for metabolic function, and that under such conditions any flux change can only be brought about by changes in protein levels, not enzyme saturation. This conclusion is supported by the fact that most glycolytic enzymes reach the highest saturation above $D = 0.5\text{h}^{-1}$ (Goel et

al, 2015), and that proteomics data showed genome-wide protein reallocation when growth rate increased beyond that point (Appendix Fig S2)."

6. This manuscript relies heavily on data from the authors' previous publications (PMID 25828364, PMID 30630406), which of course is completely fine. However, I believe this manuscript would benefit from a clarifying paragraph, in which the authors outline which of the results presented here confirm their previous findings, and which results constitute novel insights/interpretations.

Thank you for this constructive comment! We have added in the first three paragraphs in the Discussion section to clearly state which results were presented before and which results are novel in our paper.

Thank you again for sending us your revised manuscript. We have now heard back from reviewer #3 who was asked to evaluate your revised study. As you will see below, the reviewer is satisfied with the modifications made and thinks that the study is now suited for publication.

Before we can formally accept the manuscript for publication, we would ask you to address a few remaining editorial issues listed below.

REFEREE REPORTS

Reviewer #3:

I thank the authors for adequately addressing all of my concerns in this revised manuscript. I have no further concerns or comments.

The authors have made all requested editorial changes.

Accepted

1st Mar 2021

Thank you again for sending us your revised manuscript. We are now satisfied with the modifications made and I am pleased to inform you that your paper has been accepted for publication.

Corresponding Author Name: Bas Teusink, Jens Nielsen

Manuscript Number: MSB-2020-10093